# Role of multi-layer tissue composition of musculoskeletal extremities for prediction of in vivo surface indentation response and layer deformations

**Erica E. Neumann**[1,2]**, Sean Doherty**[1,2]**, James Bena**[3]**, Ahmet Erdemir**[1,2]*****

**1** Department of Biomedical Engineering, Lerner Research Institute, Cleveland Clinic, Cleveland, OH, United States of America, **2** Computational Biomodeling (CoBi) Core, Lerner Research Institute, Cleveland Clinic, Cleveland, OH, United States of America, **3** Quantitative Health Sciences, Cleveland Clinic, Cleveland, OH, United States of America

* erdemira@ccf.org

## Abstract

Emergent mechanics of musculoskeletal extremities (surface indentation stiffness and tissue deformation characteristics) depend on the underlying composition and mechanics of each soft tissue layer (i.e. skin, fat, and muscle). Limited experimental studies have been performed to explore the layer specific relationships that contribute to the surface indentation response. The goal of this study was to examine through statistical modeling how the soft tissue architecture contributed to the aggregate mechanical surface response across 8 different sites of the upper and lower extremities. A publicly available dataset was used to examine the relationship of soft tissue thickness (fat and muscle) to bulk tissue surface compliance. Models required only initial tissue layer thicknesses, making them usable in the future with only a static ultrasound image. Two physics inspired models (series of linear springs), which allowed reduced statistical representations (combined locations and location specific), were explored to determine the best predictability of surface compliance and later individual layer deformations. When considering the predictability of the experimental surface compliance, the physics inspired combined locations model showed an improvement over the location specific model (percent difference of 25.4 +/- 27.9% and 29.7 +/- 31.8% for the combined locations and location specific models, respectively). While the statistical models presented in this study show that tissue compliance relies on the individual layer thicknesses, it is clear that there are other variables that need to be accounted for to improve the model. In addition, the individual layer deformations of fat and muscle tissues can be predicted reasonably well with the physics inspired models, however additional parameters may improve the robustness of the model outcomes, specifically in regard to capturing subject specificity.

**Data Availability Statement:** Raw data is available at https://multisbeta.stanford.edu/ (doi.org/10.18735/S5R97F). Aggregated data for this study can be found at the Downloads section of the

project website (https://simtk.org/projects/multis) or directly at https://simtk.org/frs/?group_id=1032.

**Funding:** This study has been supported by United States Army Medical Research and Material Command, Department of Defense (W81XWH-15-1-0232, PI: Erdemir). The funders had no role in study design, data collection and analysis, decision to publish, or preparation of the manuscript. The views, opinions and/or findings contained in this document are those of the authors and do not necessarily reflect the views of the funding agency.

**Competing interests:** The authors have declared that no competing interests exist.

## Introduction

Studies of musculoskeletal extremity region tissue remains important, as injury to these regions is common and the tissue layers interface with prosthetics and exo-skeleton devices [1–3]. Upper and lower limbs are the two most frequently injured regions during car crashes [4]. The extremities were also more frequently wounded than other regions during recent military combat, as the extremities are difficult to protect without reduction to mobility [5]. Understanding how soft tissue behaves under loading can provide a valuable insight into several areas of biomedical research. Mechanics of musculoskeletal soft tissue layers, e.g., indentation response both at the surface and in internal tissue structures, may be utilized to identify device-tissue interactions to improve device design [6], detect soft tissue damage [7], or build accurate haptic feedback surgical simulations [8]. Improving on surgical simulations by enhancing patient realism can reduce patient exposure to inexperienced medical practitioners, as novice surgeons can gain experience in a virtual setting [9].

The bulk soft tissue surface response is dependent on the underlying tissue properties, geometry, and interactions within the composite layers of skin, fat, and muscle. Indentation response inherits viscoelastic and nonlinear behavior of underlying tissues, which are typically modeled with elaborate constitutive relationships at the material level, e.g. hyperelastic. However, this work focuses on linearized response, which still accommodates to account for major contributions of tissue layers to said response. This linear approximation has been made previously in literature for generation of simple but useful and relevant mechanical behavior estimation [10]. Additionally, this simplicity lends itself to an easier to compare and understand metric of relative indentation mechanics. The literature has reported differences of indentation response across various locations of the body [11–13], however the identification of what contributes to these differences is rarely explored in detail. Understanding of what contributes to the difference in mechanics can provide a method for modeling mechanical behavior as a function of tissue composition, which can enable future predictions of indentation response.

While individual layer composition and/or mechanics likely explain the bulk surface indentation response of soft tissue, there are limited experimental studies that investigate the mechanistic relationships of the underlying tissue layers and their respective interactions. Much of the literature examines each layer (skin, fat, or muscle) individually [14–16], in a targeted manner, or as a bulk tissue where all the layers are combined and treated as one [11,17,18], which in turn eliminates the mechanical interaction between layers that may explain the resulting mechanical behavior on a layer specific level.

The composition of layers that make up the soft tissue may lead to a different response to external forces, which has been shown for several biological tissues, including arterial and skin tissues [19–21]. Other studies have utilized finite element analysis to examine the indentation response of individual soft tissue layers [22,23], however they require applying appropriate material models and can be computationally expensive. Identifying the appropriate material model coefficients is a challenge, and can fail to capture the inherent variability between different patients. Inverse finite element analysis offers the ability to extract patient-specific response data, but requires mechanical testing, which may not be feasible in clinical or field applications, and optimization, which leads to even more simulation iterations and time [18]. Given these drawbacks, linear statistical modeling becomes a useful tool to analyze and predict how musculoskeletal soft tissue responds to loading.

The goal of this study is to determine how individual soft tissue layer architecture affects the mechanical behavior of multi-layer tissue regions across different sites within human in-vivo extremities. Two hypotheses were investigated through statistical modeling to meet this goal: 1) in vivo indentation stiffness, an aggregate mechanical behavior of multi-layer tissue

regions, is related to the thickness of skin, fat, and muscle comprising the layered tissue architecture and 2) individual tissue layer (fat and muscle) mechanical behavior can be explained using only the aggregate surface mechanical response and a quantification of structure, in this case unloaded fat and muscle layer thicknesses. These model features can easily be annotated and extracted from a single ultrasound image, making the model usable without knowledge of indentation force-displacement response for new data.

## Methods

### Dataset overview

A publicly available dataset was used for the analysis performed in this paper [24]. Briefly, indentation data was collected using an instrumented ultrasound device [25], where manual indentation was performed on 100 subjects. Informed written consent was obtained from research subjects. Data collection methods were approved by the Cleveland Clinic institutional review board under IRB # 14–1597. De-identified dissemination of data did not fall under human subjects research under Stanford University IRB # 34361.

Three important factors were used for determining indentation location: arm vs. leg, above vs. below the hinge joint, and anterior vs. posterior for the region. Eight locations then result from these criteria: the upper and lower right arms and legs in the anterior and posterior central regions. The central region was chosen for each of these areas for consistency, but also as a way to avoid indentation very close to bone which may influence mechanical behavior. The instrumented ultrasound device provided the force-displacement response over the course of an indentation trial.

Complete data were available from 95 subjects (47 male, 48 female), while data from the 5 remaining test subjects were excluded due to errors in force data collection. The force-displacement curve provided a model response variable, and ultrasound images were processed for model inputs. Ultrasound image analysis was performed by manually selecting tissue boundaries (superficial skin, skin/fat interface, fat/muscle interface, and muscle/bone interface) throughout the manual indentation procedure. The skin layer was not used in the model due to the limited resolution of the ultrasound images. Previous work analyzed how tissue response was predicted by indentation region or other demographic factors [26]. Similar to this prior work, the aggregate tissue mechanical behavior was characterized using a linear fit to the pressure vs. probe displacement data extracted during manual indentation. Pressure is defined as the force magnitude divided by the ultrasound probe contact area and displacement corresponds to the bulk tissue thickness change (measured from the superficial skin boundary to the bone boundary). Force channels were reset to zero at zero displacement and used to calculate force magnitude.

The linear fit was performed using numpy.linalg.lstsq in Python version 2.7 (http://www.python.org), where the slope of the line is equivalent to the tissue surface stiffness (note that the y-intercept was set to zero). In following, tissue compliance was calculated as the inverse of tissue surface stiffness would be tissue compliance, which is the metric of interest for this work facilitating development of physics inspired statistical representation of the data (see below). Fig 1 provides a visualization of the relationship between the experimental compliance and the tissue layer thicknesses. It can be observed that there is some clustering of location by muscle thickness. Compliance and tissue thickness appear weakly correlated for all regions of the musculoskeletal extremities, creating a need for adding musculoskeletal region as an input feature.

### Statistical tests

A linear mixed effect analysis was performed in R version 3.6.2 [27] using the 'lme4' and lmerTest' packages [28,29] to examine the relationship of muscle and fat tissue thickness on

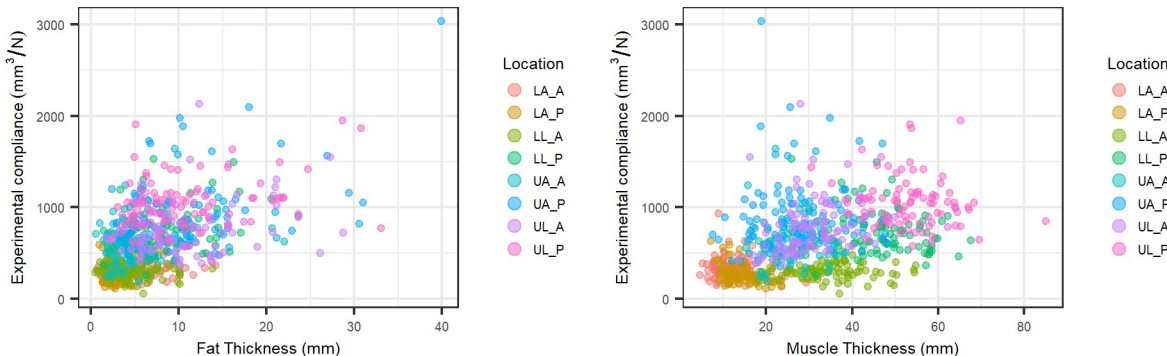

**Fig 1.** Experimental compliance vs. tissue thickness (fat–left, muscle–right). Abbreviations: LA_A–lower arm anterior, LA_P–lower arm posterior, LL_A–lower leg anterior, LL_P–lower leg posterior, UA_A–upper arm anterior, UA_P–upper arm posterior, UL_A–upper leg anterior, UL_P–upper leg posterior.

the aggregate tissue compliance. The subject ID and location were defined as random effects, while the fixed effects were initial muscle thickness and initial fat thickness. Random effects are grouping effects within which observations are expected to be correlated, while fixed effects can be thought as factors that will have a predictable effect across the population, and are the primary focus of the analysis [30]. Compliance was used as the response variable. Additionally, a linear model, using the 'stats' package [27], was used to examine the location specific relationship of muscle and fat layer thicknesses on the aggregate tissue compliance. The significance level was set to 0.05 for all model coefficients.

## Physics inspired statistical models

The physics inspired models (combined locations vs location specific) were constructed to have physical intuition. In this model, coefficients can be expressed as physically meaningful material parameters of the different tissue layers. A common physics representation of deformation is a mass-spring system, where springs resist the force applied on a mass. This work proposes that muscle and fat tissue layers can be modeled as two serial springs given the stacking of tissue layers on top of each other. These springs are attached to a rigid interface (bone) that does not move, and experience force from the movement of an ultrasound probe. Linear springs in a serial arrangement can be approximated as a single Hookean spring where the aggregate tissue spring constant ($k$) is a function of the muscle ($k_m$) and fat ($k_f$) spring constants (Eq 1).

$$\frac{1}{k} = \frac{1}{k_m} + \frac{1}{k_f} \tag{1}$$

Eq 1 is now rewritten based on substituting Hooke's Law into the definition of Young's Modulus, so that the layer spring constants are a function of the material modulus ($E$), indenter area ($A$), and initial thickness of the tissue ($t$) (Eq 2).

$$k_m = \frac{E_m A_m}{t_m}, \quad k_f = \frac{E_f A_f}{t_f} \tag{2}$$

Substituting both of the functions from Eq 2 into Eq 1, Eq 1 can then be written as:

$$\frac{1}{k} = \frac{t_m}{E_m A_m} + \frac{t_f}{E_f A_f} \tag{3}$$

Assuming that the indenter area is constant across the depth of the tissue ($A = A_m = A_f$), further simplifies Eq 3 to:

$$\frac{A}{k} = \frac{t_m}{E_m} + \frac{t_f}{E_f} \qquad (4)$$

The model response variable, ($A/k$) is the aggregate compliance in units of mm$^3$/N. The fixed effects include the thickness of muscle ($t_m$) and fat ($t_f$), respectively in mm. The random effects used when all locations are combined were location and subject ID, however location specific models were also developed for the physics-based method. The resulting coefficients of the linear model represent the inverse modulus (1/MPa) of the muscle ($1/E_m$) and fat ($1/E_f$) layers, respectively. This spring based formulation has value in providing easier to compare metrics, while also allowing for understandable differences in the coefficients for muscle and fat. The coefficients could be useful in predicting smaller deformations, as with larger deformations the hyperbolic nature of tissue stiffness will cause these models to under predict aggregate stiffness. Given this, the statistical representation did not include an intercept (i.e. the model is forced through the (x, y) point of (0, 0)). This is an important feature for the physics inspired model, as the slope of this fit is proportional to the inverse of Young's modulus, a measure of material response. Given this, it is important for the model to intersect (0, 0) since with zero stress there should be zero strain. These model coefficients be utilized to predict layer response for a given loading as described below.

## Tissue deformations: Physics inspired model vs experimental

Utilizing the physically meaningful parameters extracted from the physics inspired models, model results were compared to the experimental results for each tissue layer. The model coefficients, $1/E_m$ and $1/E_f$, were used to calculate the deformation of the muscle and fat layers (Eq 5).

$$\Delta t_{m,pred} = \frac{t_m}{E_m} * \frac{F_{max}}{A}, \qquad \Delta t_{f,pred} = \frac{t_f}{E_f} * \frac{F_{max}}{A} \qquad (5)$$

The predicted displacements, $\Delta t_{m,pred}$ and $\Delta t_{f,pred}$, are functions of layer thickness ($t_m$ or $t_f$), maximum applied indentation force ($F_{max}$), and indenter area ($A$). Predicted displacements were compared to the experimental displacements of the muscle and fat layers using the location specific and combined location model coefficients ($1/E_m$ and $1/E_f$).

## Conventional statistical models with intercepts for sensitivity analysis

Conventional linear statistical models were also developed, where the model intercept was included in representation. While the coefficients of this model may not have a directly interpretable relationship to layer mechanical properties (as elaborated upon above), they still provide valuable intuition. The coefficients of the intercept model can be used to consider the effect a unit change in thickness will have on tissue compliance over the range of data that was collected. The formulation is otherwise identical to the statistical tests methods used above, where a linear mixed effect model was formulated with random effects set to location and subject ID when all locations are combined. Location specific models were also generated. These two models were developed to examine the sensitivity of the physics inspired models to having the intercept removed.

## Results

The physics inspired model coefficients varied across locations and showed significance for all 8 indentation locations for both the fat and muscle coefficients (Table 1). The absolute percent difference between the predicted compliance and experimental compliance was 29.7 +/- 31.8%

**Table 1. Physics inspired model coefficients (mm$^3$/N) for each location.**

|  | LA_A | LA_P | LL_A | LL_P | UA_A | UA_P | UL_A | UL_P | Combined |
|---|---|---|---|---|---|---|---|---|---|
| Muscle | 13.36*** (2.02) | 8.49*** (1.66) | 6.87*** (0.61) | 10.49*** (1.06) | 17.03*** (1.36) | 18.75*** (1.90) | 18.85*** (1.88) | 14.67*** (1.03) | 3.77** (1.14) |
| Fat | 26.94*** (4.10) | 44.96*** (8.00) | 12.29** (3.98) | 27.82*** (5.39) | 27.21*** (5.95) | 38.13*** (4.42) | 17.52*** (4.766) | 23.31*** (4.39) | 18.10*** (2.03) |

Standard errors are reported in parentheses below the coefficient estimate. Abbreviations: LA_A–lower arm anterior, LA_P–lower arm posterior, LL_A–lower leg anterior, LL_P–lower leg posterior, UA_A–upper arm anterior, UA_P–upper arm posterior, UL_A–upper leg anterior, UL_P–upper leg posterior. Asterisks represent statistical significance, where ***<0.001, **<0.01, *<0.05.

and 25.4 +/- 27.9% for the location specific and combined location physics-based models, respectively. Fig 2 shows correlation plots for the experimental vs. predicted compliance of both the location specific and combined physics-based models.

The conventional model (with intercept) coefficients varied across locations and showed significance in 2 locations for the muscle coefficient and 5 locations for the fat coefficient (Table 2). The coefficients for the combined locations were all significant. The absolute percent difference between the predicted compliance and experimental compliance was 26.8 +/- 29.8% and 25.4 +/- 28.0% for the location specific and combined location intercept models, respectively. Fig 3 shows correlation plots for the experimental vs. predicted compliance of both the location specific and combined models.

A comparison of experimental to predicted fat displacement is provided for the four different model coefficients to evaluate the second hypothesis of this work. Predicted fat displacement was calculated using the physics inspired, location specific model coefficients (Fig 4) and combined location model coefficients (Fig 5).

Similarly, experimental muscle displacement comparison to predicted muscle displacement are shown for the physics inspired, location specific model coefficients (Fig 6) and combined location model coefficients (Fig 7).

## Discussion

The initial exploration of the data (Fig 1) highlights the region specific and individualized nature of tissue stiffness. Experimental compliance and tissue thickness show a weak positive

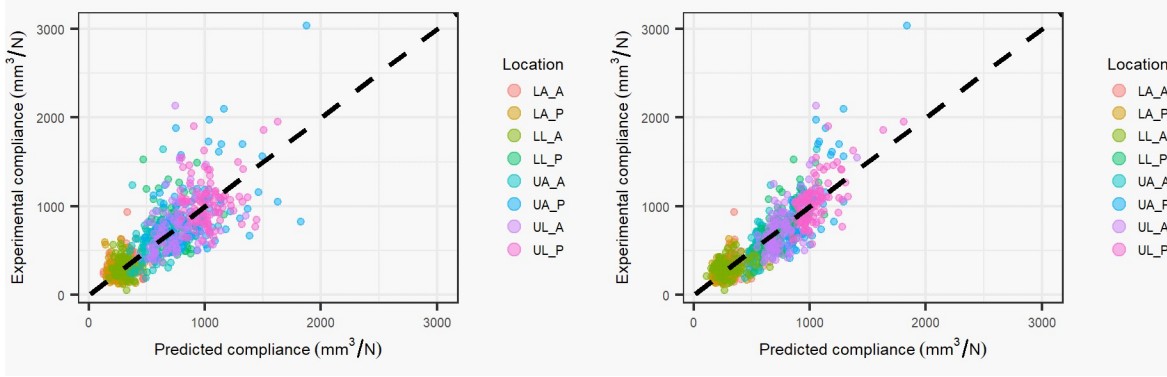

**Fig 2. Physics inspired model: Experimental compliance vs. predicted compliance (black dashed line equation: y = x).** The left plot shows results using location specific coefficients and the right plot shows results using coefficients from all locations combined. Abbreviations: LA_A–lower arm anterior, LA_P–lower arm posterior, LL_A–lower leg anterior, LL_P–lower leg posterior, UA_A–upper arm anterior, UA_P–upper arm posterior, UL_A–upper leg anterior, UL_P–upper leg posterior.

**Table 2. Coefficients of conventional statistical models (with intercept) for location specific and combined location models.**

|  | LA_A | LA_P | LL_A | LL_P | UA_A | UA_P | UL_A | UL_P | Combined |
|---|---|---|---|---|---|---|---|---|---|
| Intercept | 288.3*** (45.7) | 365.5*** (47.0) | 338.3*** (51.8) | 803.3*** (158.7) | 428.3*** (101.9) | 415.9** (155.5) | 324.0* (134.6) | 1032.4*** (197.5) | 395.4*** (88.6) |
| Muscle | -1.24 (2.87) | -7.64** (2.44) | -0.49 (1.24) | -3.16 (2.86) | 3.55 (3.44) | 8.47 (4.26) | 10.06* (4.09) | -2.01 (3.31) | 3.30** (1.16) |
| Fat | 10.61* (4.30) | 7.79 (7.85) | -1.79 (3.95) | 9.46 (6.00) | 15.51* (6.15) | 27.93*** (5.73) | 12.98* (5.02) | 11.95** (4.43) | 17.71*** (2.04) |

Standard errors are reported in parentheses below the coefficient estimate. Abbreviations: LA_A–lower arm anterior, LA_P–lower arm posterior, LL_A–lower leg anterior, LL_P–lower leg posterior, UA_A–upper arm anterior, UA_P–upper arm posterior, UL_A–upper leg anterior, UL_P–upper leg posterior. Asterisks represent statistical significance, where ***<0.001, **<0.01, *<0.05.

correlation for both fat and muscle. This figure also highlights how the distribution of muscle thickness is noticeably clustered by location, which is expected given that the lower extremities account for a larger percentage of muscle mass relative to the upper extremities [31]. This relationship holds for skeletal muscle partial volume as well [31]. It can be seen in both Tables 1 and 2 in the combined region column that the fat modulus was close to 5 times more compliant than the muscle's modulus. These coefficients appear to do a reasonable job at estimating tissue aggregate compliance, with mean error slightly lower than 30%. Using fat and muscle thickness as the only fixed effects in a model to predict tissue response could be valuable based on how accurate of mechanical response is needed. For instance, a diabetic foot showed nearly a 100% change in Young's Modulus compared to a healthy control [32]. This suggests that 30% error may be sufficient for other pathologies but this is likely highly disease specific. Other factors can also not be accounted for in this model, as Zheng and Mak saw a 460% change in effective modulus based on patient posture change, while the thickness only changed 10% in the lower leg region [13]. A linear model would fail to capture this dramatic increase in modulus since the thickness change was quite small. In relation to the first hypothesis (that layer thickness is predictive of mechanics), fat and muscle thicknesses provide a rough approximation of tissue response in a resting position, but more complicated scenarios or higher accuracy may need different model formulation.

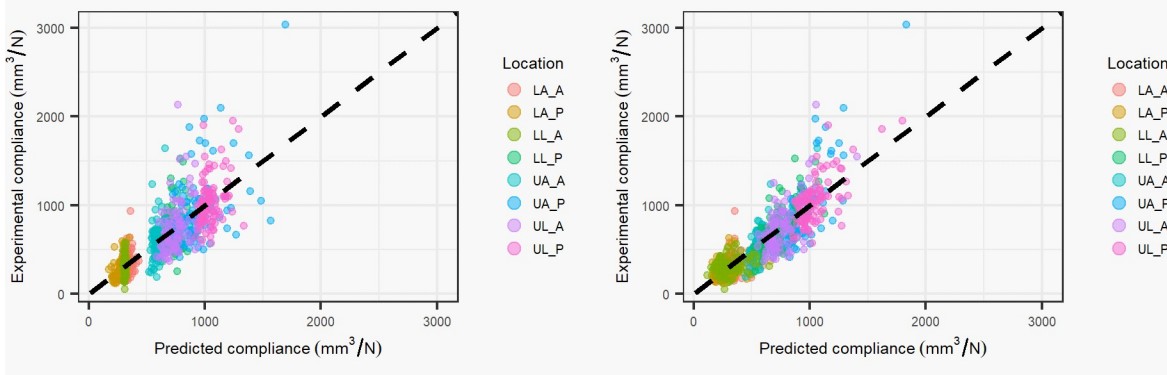

**Fig 3. Conventional statistics model: Experimental compliance vs. predicted compliance (black dashed line equation: y = x).** The left plot shows results using location specific coefficients and the right plot shows results using coefficients from all locations combined. Abbreviations: LA_A–lower arm anterior, LA_P–lower arm posterior, LL_A–lower leg anterior, LL_P–lower leg posterior, UA_A–upper arm anterior, UA_P–upper arm posterior, UL_A–upper leg anterior, UL_P–upper leg posterior.

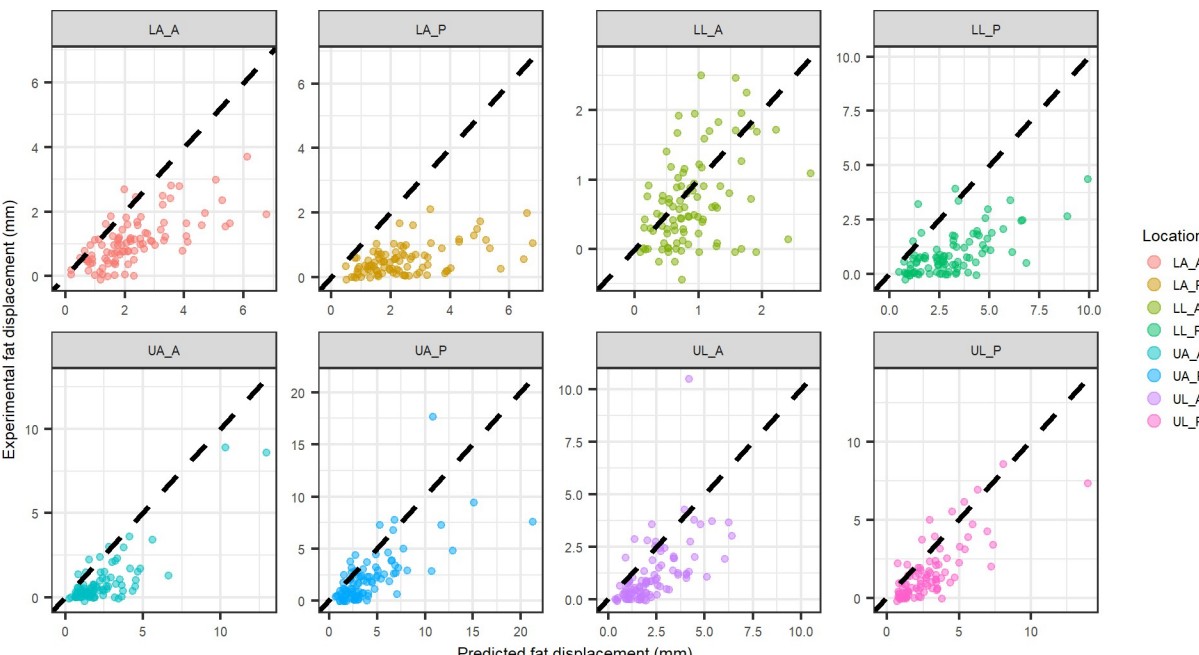

**Fig 4. Physics inspired model displacement vs. experimental displacement for fat tissue at each indentation location using coefficients from each individual location (black dashed line equation: y = x).** Abbreviations: LA_A–lower arm anterior, LA_P–lower arm posterior, LL_A–lower leg anterior, LL_P–lower leg posterior, UA_A–upper arm anterior, UA_P–upper arm posterior, UL_A–upper leg anterior, UL_P–upper leg posterior.

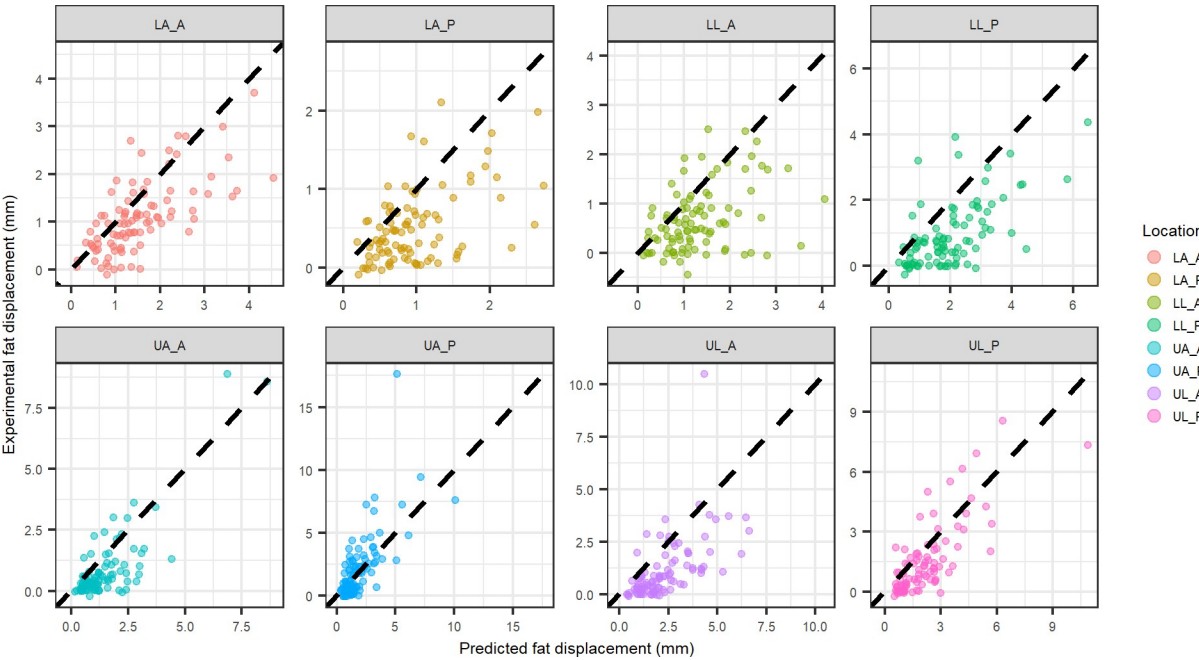

**Fig 5. Physics inspired model displacement vs. experimental displacement for fat tissue at each indentation location using coefficients from all locations combined (black dashed line equation: y = x).** Abbreviations: LA_A–lower arm anterior, LA_P–lower arm posterior, LL_A–lower leg anterior, LL_P–lower leg posterior, UA_A–upper arm anterior, UA_P–upper arm posterior, UL_A–upper leg anterior, UL_P–upper leg posterior.

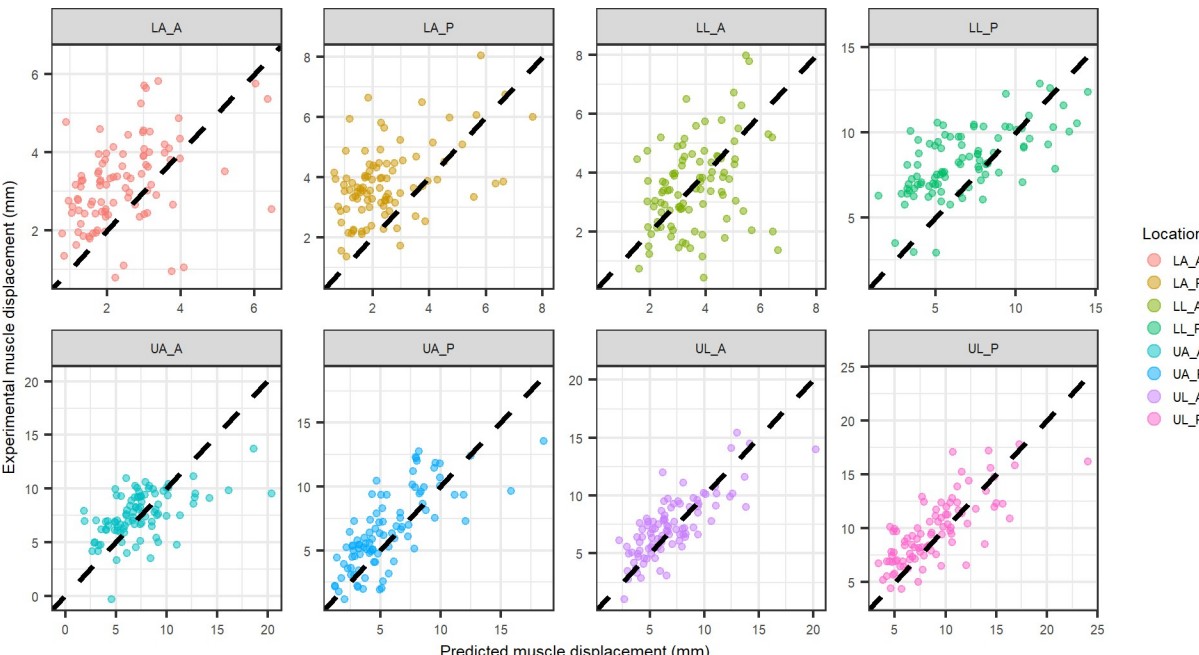

**Fig 6. Physics inspired model displacement vs. experimental displacement for muscle tissue at each indentation location using coefficients from each individual location (black dashed line equation: y = x).** Abbreviations: LA_A–lower arm anterior, LA_P–lower arm posterior, LL_A–lower leg anterior, LL_P–lower leg posterior, UA_A–upper arm anterior, UA_P–upper arm posterior, UL_A–upper leg anterior, UL_P–upper leg posterior.

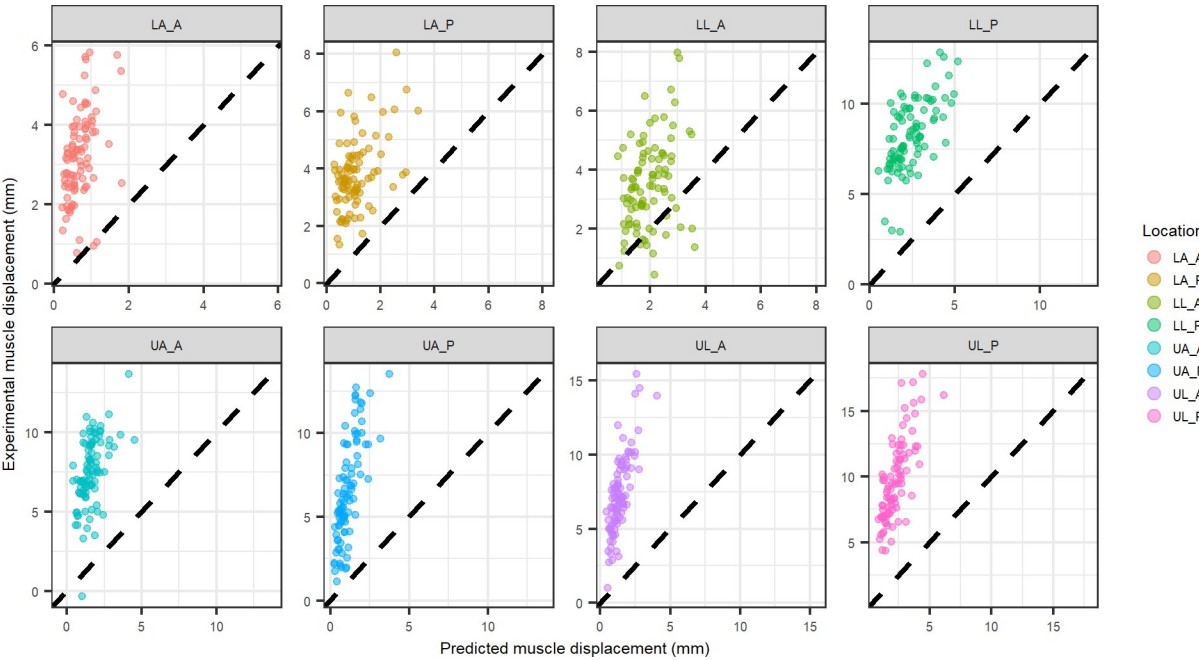

**Fig 7. Physics inspired model displacement vs. experimental displacement for muscle tissue at each indentation location using coefficients from all locations combined (black dashed line equation: y = x).** Abbreviations: LA_A–lower arm anterior, LA_P–lower arm posterior, LL_A–lower leg anterior, LL_P–lower leg posterior, UA_A–upper arm anterior, UA_P–upper arm posterior, UL_A–upper leg anterior, UL_P–upper leg posterior.

Improved fit to the experimental compliance data was shown when combining all the locations (Figs 2 & 3), in comparison to fitting locations separately (4.3% improvement in mean percent difference for the physics model, 1.4% improvement in mean percent difference for the intercept based model). This suggest that subject variability is an important factor in modeling of aggregate stiffness, with an improvement of 4.3% in mean percent difference when subject effects are considered, as subject effects are not a factor in the location specific models given that each patient only has one measurement for each location of indentation. While the intercept coefficient was significant in the intercept combined location model, the fat and muscle coefficients were similar to the physics based combined location model. A percent difference of 13.29% was observed for the muscle coefficient and a percent difference of 2.17% between the fat coefficients. This suggests that the intercept coefficient for the combined model is small enough to not contribute appreciably to the prediction of aggregate tissue compliance.

The layer specific mechanics predictions using the location specific physics model compared well to the experimental layer deformation of muscle (UA_P, UL_P, UL_A) in some cases but poorly in others (Fig 6). Additionally, when looking at the comparison of the predicted deformation to the experimental deformation for the all locations combined physics model (Fig 7), the muscle deformation is significantly under-predicted. Fat tends to be over predicted for the combined locations model (Fig 4), suggesting the under prediction of muscle deformation is compensated by an over prediction for fat deformation. The location specific coefficients were larger in magnitude (Table 2), resulting in better correlations with the experimental deformation (Fig 6). However, when comparing the experimental fat layer deformations, the combined location physics model exhibited better predictability (Fig 5). There appears to be more clustering of the muscle thickness based on location, whereas fat thickness is more distributed within various locations. Since the intercept was forced to zero for the physics based model, the clustering of the muscle likely leads to the improved fit of the location specific model when examining muscle specific mechanics.

A main limitation of this study is the inability for the ultrasound to capture the indentation response of the skin layer, due to poor resolution in comparison to the small skin thickness. The skin layer was therefore ignored from the statistical models. A previous FE study showed that variations in the skin modulus contributed most to the forearm mechanical response [33], however since only very small, if any, displacement was recorded for the skin layer, this dataset is unable to distinguish the skin layer contribution to the surface indentation response. It was also observed that skin thickness is similar across the 8 indentation sites ($< 1$ mm range of mean thickness), further suggesting that skin thickness may not be contributing to the variation in mechanical surface response of the soft tissue. Higher resolution ultrasound images may be necessary for future work to include skin parameters within models. In addition, the statistical models that were used in this study were not designed to include patient specific or location specific material properties, i.e. assumed fat and muscle mechanical properties will be similar among participants. Finally, the experimental indentation response was assumed to be linear. Previous work showed that this was an appropriate assumption for most cases [26], however, future studies may benefit from quantifying the non-linear behavior of the surface indentation response. This would allow these models to be more accurate in predicting large deformations.

## Conclusion

This study revealed several important considerations when predicting the mechanical behavior of layered tissue structures using statistical models. Our first hypothesis was supported by

statistical significance of coefficients for both fat and muscle in combined location models and for location-specific models of the physics inspired representation. The statistical models presented in this work allow for a rough estimate of the fat and muscle displacement based on the initial thickness of each respective tissue. However, depending on the tissue of interest (fat vs muscle) or the biomechanical marker (tissue deformation vs indentation compliance) one may want to utilize different types of physics inspired models (location specific or combined location). Our results indicate that tissue thickness affects emergent mechanical behavior of musculoskeletal tissue layers. However, it is also clear that the thickness alone is not enough to fully describe the aggregate surface compliance and accurately predict individual layer deformations. Direct measurement of mechanics can assist development and calibration of individualized finite element representations of musculoskeletal extremities. Without such measurements statistical models (as derived in this study) provide the means to achieve individualized predictions. Nonetheless, more comprehensive statistical models or data-driven approaches such as machine learning may yield improved accuracy. Future work should consider additional variables, such as demographics, related to patient specific response.

## Author Contributions

**Conceptualization:** Erica E. Neumann, Ahmet Erdemir.

**Data curation:** Erica E. Neumann.

**Formal analysis:** Erica E. Neumann, James Bena.

**Funding acquisition:** Ahmet Erdemir.

**Investigation:** Erica E. Neumann.

**Methodology:** Erica E. Neumann, James Bena.

**Software:** Erica E. Neumann, Sean Doherty.

**Supervision:** Ahmet Erdemir.

**Visualization:** Erica E. Neumann, Sean Doherty.

**Writing – original draft:** Erica E. Neumann.

**Writing – review & editing:** Sean Doherty, James Bena, Ahmet Erdemir.

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
