## [Decision Letter · Decision Letter 0]

19 Oct 2022

PONE-D-22-21815Role of multi-layer tissue composition of musculoskeletal extremities for prediction of in vivo surface indentation response and layer deformationsPLOS ONE

Dear Dr. Erdemir,

Thank you for submitting your manuscript to PLOS ONE. After careful consideration, we feel that it has merit but does not fully meet PLOS ONE’s publication criteria as it currently stands. Therefore, we invite you to submit a revised version of the manuscript that addresses the points raised during the review process.

We look forward to receiving your revised manuscript.

Kind regards,

Yaodong Gu

Academic Editor

PLOS ONE

Journal Requirements:

a) Did participants provide their written or verbal informed consent to participate in this study?

"This study has been supported by USAMRMC, DoD (W81XWH-15-1-0232, PI: Erdemir)"

"This study has been supported by United States Army Medical Research and Material Command, Department of Defense (W81XWH-15-1-0232, PI: Erdemir). The views, opinions and/or findings contained in this document are those of the authors and do not necessarily reflect the views of the funding agency. "

5. Please ensure that you refer to Figures 6 and 7 in your text as, if accepted, production will need this reference to link the reader to the figure.

Additional Editor Comments:

Just check the minor questions raised by reviewer 2

Reviewers' comments:

Reviewer's Responses to Questions

**Comments to the Author**

1. Is the manuscript technically sound, and do the data support the conclusions?

Reviewer #1: Yes

Reviewer #2: Yes

2. Has the statistical analysis been performed appropriately and rigorously? 

Reviewer #1: Yes

Reviewer #2: Yes

3. Have the authors made all data underlying the findings in their manuscript fully available?

Reviewer #1: No

Reviewer #2: Yes

4. Is the manuscript presented in an intelligible fashion and written in standard English?

Reviewer #1: Yes

Reviewer #2: Yes

5. Review Comments to the Author

Reviewer #2: Review comment

This manuscript entitled “Role of multi-layer tissue composition of musculoskeletal extremities for prediction of in vivo surface indentation response and layer deformations” primarily aimed to determine how individual soft tissue layer architecture affects the mechanical behavior of multi-layer tissue regions across different sites within human in-vivo extremities. The authors bring an interesting study, but there are still some problems that cannot up this study to a publishing level. Some suggestions are listed in the specific comments below.

Specific comments:

The current introduction section provides limited background information, please further highlight the research necessity and potential value. Why is it important to determine the effects of individual soft tissue layer architecture on the mechanical behavior of multi-layer tissue regions across different sites within human in-vivo extremities.

What was the basis for selecting the eight sites of the upper and lower extremities?

The methodology involving technical subjects (especially physics inspired statistical models) is difficult to understand, please provide more details to make it clearer.

‘…that the upper leg region is very likely to have a thicker layer of muscle than the lower arm region for most individuals.’, please add a ref here.

In summary, please ensure that your manuscript is prepared correctly (without any grammatical and spelling mistakes) and formatted before submitting a revision.

6. PLOS authors have the option to publish the peer review history of their article (what does this mean?). If published, this will include your full peer review and any attached files.

Reviewer #1: No

Reviewer #2: No

---

## [Author Response · Author response to Decision Letter 0]

21 Nov 2022

Response to the Reviewer:

We thank the reviewer for their valuable feedback. Responses to the comments have been included below, with reference line numbers in the marked up manuscript. 

Reviewer Specific Comments to the Corresponding Author:

1. The current introduction section provides limited background information, please further highlight the research necessity and potential value. Why is it important to determine the effects of individual soft tissue layer architecture on the mechanical behavior of multi-layer tissue regions across different sites within human in-vivo extremities.

Author Response: 

The introduction section was expanded with some of the research that justifies the statistical modeling effort. Several additional citations and research avenues were added in lines 44-49. Surgical simulation represents one of the most intriguing areas (lines 52-54). Justification on why statistical models were used over traditional physics simulations like finite element analysis was also added to the introduction. 

2. What was the basis for selecting the eight sites of the upper and lower extremities?

Author Response: 

 The paper was revised to justify why the 8 different locations were selected in lines 95-101. While anatomical measurements were performed in reference 20 at 48 different locations to obtain tissue thicknesses across the musculoskeletal extremities, the central region of each extremity was chosen given its distance from the bone. Proximity to an extremity bone, such as a knee, made reproducible stiffness measurements more challenging. 

3. The methodology involving technical subjects (especially physics inspired statistical models) is difficult to understand, please provide more details to make it clearer.

Author Response: 

 The physics inspired statistical model description was updated in lines 134-162. Greater effort was put in justifying the physics rational of the model. Further explanation for Equation 1, 2, and 3 was provided. Further justification was provided on why the model was forced through a 0 y-intercept in lines 159-163. A sentence was added into the statistical tests section to provide a quick explanation on how linear mixed effect models function (lines 129-131). 

4. ‘…that the upper leg region is very likely to have a thicker layer of muscle than the lower arm region for most individuals.’, please add a ref here.

Author Response: 

 A citation from a previous study has been added to support this point with reference 27 in the updated bibliography. Muscle in the thigh tends to outweigh muscle in the arm, and the lower extremities exceed the upper extremities by partial volume. Lines 260-263 have been updated with this study’s work. 

5. In summary, please ensure that your manuscript is prepared correctly (without any grammatical and spelling mistakes) and formatted before submitting a revision.

Author Response: 

 Various edits and corrections were made throughout the paper to improve the paper outside the above comments, such as removing first person language.

---

## [Decision Letter · Decision Letter 1]

5 Dec 2022

PONE-D-22-21815R1Role of multi-layer tissue composition of musculoskeletal extremities for prediction of in vivo surface indentation response and layer deformationsPLOS ONE

Dear Dr. Erdemir,

Thank you for submitting your manuscript to PLOS ONE. After careful consideration, we feel that it has merit but does not fully meet PLOS ONE’s publication criteria as it currently stands. Therefore, we invite you to submit a revised version of the manuscript that addresses the points raised during the review process.

We look forward to receiving your revised manuscript.

Kind regards,

Yaodong Gu

Academic Editor

PLOS ONE

Journal Requirements:

Reviewers' comments:

Reviewer's Responses to Questions

**Comments to the Author**

1. If the authors have adequately addressed your comments raised in a previous round of review and you feel that this manuscript is now acceptable for publication, you may indicate that here to bypass the “Comments to the Author” section, enter your conflict of interest statement in the “Confidential to Editor” section, and submit your "Accept" recommendation.

Reviewer #1: (No Response)

Reviewer #2: All comments have been addressed

2. Is the manuscript technically sound, and do the data support the conclusions?

Reviewer #1: Partly

Reviewer #2: Yes

3. Has the statistical analysis been performed appropriately and rigorously? 

Reviewer #1: I Don't Know

Reviewer #2: Yes

4. Have the authors made all data underlying the findings in their manuscript fully available?

Reviewer #1: No

Reviewer #2: Yes

5. Is the manuscript presented in an intelligible fashion and written in standard English?

Reviewer #1: Yes

Reviewer #2: Yes

6. Review Comments to the Author

Reviewer #1: The author did not solve the comments raised by the reviewer in the previous round.

1. This manuscript examines how the soft tissue architecture contributed to the aggregate mechanical surface response across 8 different sites of the upper and lower extremities. There is a great insight into investigating the soft tissue. However, the abstract section and the results section mentioned that ‘the fat layer deformation was predicted best by the combined locations model, while the muscle layer deformation was predicted best by the location-specific model’, it’s not in line with the purpose presented in the paper, and the reviewer is confused as to whether the authors intend to compare the prediction effects of the two prediction models on soft tissue deformation or to prove how the soft tissue architecture contributed to the aggregate mechanical surface response across 8 different sites of the upper and lower extremities? The title and purpose were more inclined to explore the role of multi-layer tissue composition of musculoskeletal extremities for the prediction of in vivo surface indentation response and layer deformations. But in the suggestion section, the results are more inclined to compare the difference between the two prediction models.

2. The methods section of this manuscript is more important, the deformation simulation of soft tissue is an important part of calculated results, but the existing deformation calculation methods and the physical models are difficult to achieve high accuracy and real-time performance. Therefore, the author how to make sure the accuracy and applicability of the calculation method in the article? Is there any relevant literature reference?

3. The complex mechanical properties of human soft tissue structure are closely related to physiological and pathological states, so it is necessary to establish and adopt a suitable constitutive model to describe the deformation behavior of soft tissue in mechanical imaging. But the construction of this model is not mentioned in the methods section of the author. Please descript it in detail.

4. “two hypotheses were investigated to meet this goal”, but in the suggestion section, the reviewer did not see the author make a detailed discussion on hypothesis 2.

5. Fat tends to be over-predicted for the combined locations model suggesting the underprediction of muscle deformation is compensated by an over-prediction for fat deformation. Can specific analyses or quantitative data be used to demonstrate the interaction between fat and muscle deformation? Or provide relevant reference documentation.

6. As the author mentioned in the limitation part, there are big problems in the current research method, which is also the concern of the reviewer.

7. All Figure’s clarity is not enough, recommended to improve their quality. The format of the Table is wrong. There are many handwriting errors in the manuscript, such as too many spaces. Please check carefully. Figure 1 needs more comments to explain.

Reviewer #2: (No Response)

7. PLOS authors have the option to publish the peer review history of their article (what does this mean?). If published, this will include your full peer review and any attached files.

Reviewer #1: No

Reviewer #2: No

---

## [Author Response · Author response to Decision Letter 1]

16 Jan 2023

Response to the Reviewer:

We thank the reviewer for their valuable feedback. Responses to the comments have been included below, with reference line numbers in the marked up manuscript. 

Reviewer Specific Comments to the Corresponding Author:

1. This manuscript examines how the soft tissue architecture contributed to the aggregate mechanical surface response across 8 different sites of the upper and lower extremities. There is a great insight into investigating the soft tissue. However, the abstract section and the results section mentioned that ‘the fat layer deformation was predicted best by the combined locations model, while the muscle layer deformation was predicted best by the location-specific model’, it’s not in line with the purpose presented in the paper, and the reviewer is confused as to whether the authors intend to compare the prediction effects of the two prediction models on soft tissue deformation or to prove how the soft tissue architecture contributed to the aggregate mechanical surface response across 8 different sites of the upper and lower extremities? The title and purpose were more inclined to explore the role of multi-layer tissue composition of musculoskeletal extremities for the prediction of in vivo surface indentation response and layer deformations. But in the suggestion section, the results are more inclined to compare the difference between the two prediction models.

Author Response:

 A new paragraph was added at the beginning of the discussion to more closely describe the interplay between layers. The discussion section was reworked to make it more clear how the discussion section was related to the two hypotheses. The abstract was also reworked for improved paper clarity.

2. The methods section of this manuscript is more important, the deformation simulation of soft tissue is an important part of calculated results, but the existing deformation calculation methods and the physical models are difficult to achieve high accuracy and real-time performance. Therefore, the author how to make sure the accuracy and applicability of the calculation method in the article? Is there any relevant literature reference?

Author Response:

 The accuracy of the calculation method can be shown in Figure 4 and 5, showing the Physics based model has an error of around 25-30%. This was discussed in the newly added first discussion paragraph (lines 284-304). Whether this is sufficiently high accuracy would depend on the application of the models. These models requires only simple annotations of ultrasound for tissue thicknesses, which makes them usable without force transducers or camera systems. The models are probably less accurate than some complex models that exist in literature, but require only simple inputs and no parameter search. The value of the models depends on the scenario, which has been added in lines 291-301. 

3. The complex mechanical properties of human soft tissue structure are closely related to physiological and pathological states, so it is necessary to establish and adopt a suitable constitutive model to describe the deformation behavior of soft tissue in mechanical imaging. But the construction of this model is not mentioned in the methods section of the author. Please descript it in detail.

Author Response:

 This model is phenomenological spring based representation of layers. If this work used finite element analysis, it would need a constitutive model. Additionally, tissue behavior was approximated to springs in series rather than being modeled with a constitutive model such as the Mooney-Rivlin model due to a lack of time related data, such as rate dependency and accurate strain measurements across deformation. Spring based modeling is more common in mesh based applications [1], gait [2,3], or simplified modeling of extremities [4]. A benefit of this simplification is that a single metric for stiffness may allow for simpler comparison between regions and subjects as [1] and [2] note the benefit of simplification on providing understanding. Hyperelastic materials also still behave like linear elastic materials under small loads. The above response has been more clearly explained in several portions of the paper, in lines 59-66, 177-179, 299-303. 

4. “two hypotheses were investigated to meet this goal”, but in the suggestion section, the reviewer did not see the author make a detailed discussion on hypothesis 2.

Author Response:

 The discussion was reformatted to better align the discussion with the two hypotheses.

5. Fat tends to be over-predicted for the combined locations model suggesting the underprediction of muscle deformation is compensated by an over-prediction for fat deformation. Can specific analyses or quantitative data be used to demonstrate the interaction between fat and muscle deformation? Or provide relevant reference documentation.

Author Response:

 This is what we are suspecting, based on the spring based coefficients. Because the problem was formulated as an optimization to aggregate surface response, many combinations of different spring stiffnesses are possible. The springs could be fit individually, but that requires annotation of the different layers at all times. 

6. As the author mentioned in the limitation part, there are big problems in the current research method, which is also the concern of the reviewer.

Author Response:

 Some of the limitations of this work, such as a lack of inclusion of demographic related parameters in the model are being addressed in future models built. Limitations such as lack of resolution to determine skin thickness are not addressable unless different ultrasound equipment is used, which may not be feasible in a hospital setting. The standard deviations for skin was greater than the actual measurement in this dataset, so its inclusion in a model would be just fitting to noise [5]. 25% accuracy may be sufficient depending on application. 

7. All Figure’s clarity is not enough, recommended to improve their quality. The format of the Table is wrong. There are many handwriting errors in the manuscript, such as too many spaces. Please check carefully. Figure 1 needs more comments to explain.

Author Response:

 Figures were formatted with the PACE editor provided by PLOS ONE. The authors also note that the PLOS ONE compiled submission PDF includes low-resolution preview images of the figures after the reference list. The function of these previews is to allow you to download the entire submission as quickly as possible. Click the link at the top of each preview page to download a high-resolution version of each figure. Figure 1 was elaborated upon in Lines 135-139.Tables were reformatted to include all non-title elements inside the table legend. Typographical errors in the manuscript were fixed in the resubmission and the authors thank the reviewer for pointing them out. 

Bibliography:

1. Kähler K, Haber J, Seidel H-P. Geometry-based Muscle Modeling for Facial Animation. 

2. Passive Dynamic Walking - Tad McGeer, 1990. Available: https://journals.sagepub.com/doi/abs/10.1177/027836499000900206

3. Whittington BR, Thelen DG. A Simple Mass-Spring Model with Roller Feet can induce the Ground Reactions Observed in Human Walking. J Biomech Eng. 2009;131: 011013. doi:10.1115/1.3005147

4. Butler RJ, Crowell HP, Davis IM. Lower extremity stiffness: implications for performance and injury. Clin Biomech. 2003;18: 511–517. doi:10.1016/S0268-0033(03)00071-8

5. Neumann EE, Owings TM, Schimmoeller T, Nagle TF, Colbrunn RW, Landis B, et al. Reference data on thickness and mechanics of tissue layers and anthropometry of musculoskeletal extremities. Sci Data. 2018;5: 180193. doi:10.1038/sdata.2018.193

---

## [Decision Letter · Decision Letter 2]

6 Apr 2023

Role of multi-layer tissue composition of musculoskeletal extremities for prediction of in vivo surface indentation response and layer deformations

PONE-D-22-21815R2

Dear Dr. Erdemir,

We’re pleased to inform you that your manuscript has been judged scientifically suitable for publication and will be formally accepted for publication once it meets all outstanding technical requirements.

Kind regards,

Yaodong Gu

Academic Editor

PLOS ONE

Reviewers' comments:

Reviewer's Responses to Questions

**Comments to the Author**

1. If the authors have adequately addressed your comments raised in a previous round of review and you feel that this manuscript is now acceptable for publication, you may indicate that here to bypass the “Comments to the Author” section, enter your conflict of interest statement in the “Confidential to Editor” section, and submit your "Accept" recommendation.

Reviewer #1: All comments have been addressed

2. Is the manuscript technically sound, and do the data support the conclusions?

Reviewer #1: Partly

3. Has the statistical analysis been performed appropriately and rigorously? 

Reviewer #1: Yes

4. Have the authors made all data underlying the findings in their manuscript fully available?

Reviewer #1: Yes

5. Is the manuscript presented in an intelligible fashion and written in standard English?

Reviewer #1: Yes

6. Review Comments to the Author

Reviewer #1: all comments have been addressed

7. PLOS authors have the option to publish the peer review history of their article (what does this mean?). If published, this will include your full peer review and any attached files.

Reviewer #1: No

---

## [Editor Report · Acceptance letter]

14 Apr 2023

PONE-D-22-21815R2 

Role of multi-layer tissue composition of musculoskeletal extremities for prediction of in vivo surface indentation response and layer deformations 

Dear Dr. Erdemir:

I'm pleased to inform you that your manuscript has been deemed suitable for publication in PLOS ONE. Congratulations! Your manuscript is now with our production department. 

Kind regards, 

on behalf of

Professor Yaodong Gu 

Academic Editor

PLOS ONE